# Rapid Learning or Feature Reuse? Towards Understanding the Effectiveness of MAML

**Aniruddh Raghu** *
MIT
araghu@mit.edu

**Maithra Raghu** *
Cornell University & Google Brain
maithrar@gmail.com

**Samy Bengio**
Google Brain

**Oriol Vinyals**
DeepMind

## Abstract

An important research direction in machine learning has centered around developing meta-learning algorithms to tackle few-shot learning. An especially successful algorithm has been Model Agnostic Meta-Learning (MAML), a method that consists of two optimization loops, with the outer loop finding a *meta-initialization*, from which the inner loop can efficiently learn new tasks. Despite MAML's popularity, a fundamental open question remains – is the effectiveness of MAML due to the meta-initialization being primed for *rapid learning* (large, efficient changes in the representations) or due to *feature reuse*, with the meta-initialization already containing high quality features? We investigate this question, via ablation studies and analysis of the latent representations, finding that feature reuse is the dominant factor. This leads to the ANIL (Almost No Inner Loop) algorithm, a simplification of MAML where we *remove* the inner loop for all but the (task-specific) head of the underlying neural network. ANIL matches MAML's performance on benchmark few-shot image classification and RL and offers computational improvements over MAML. We further study the precise contributions of the head and body of the network, showing that performance on the test tasks is entirely determined by the quality of the learned features, and we can remove even the head of the network (the NIL algorithm). We conclude with a discussion of the rapid learning vs feature reuse question for meta-learning algorithms more broadly.

## 1 Introduction

A central problem in machine learning is *few-shot learning*, where new tasks must be learned with a very limited number of labelled datapoints. A significant body of work has looked at tackling this challenge using meta-learning approaches (Koch et al., 2015; Vinyals et al., 2016; Snell et al., 2017; Finn et al., 2017; Santoro et al., 2016; Ravi and Larochelle, 2016; Nichol and Schulman, 2018). Broadly speaking, these approaches define a family of tasks, some of which are used for training and others solely for evaluation. A proposed meta-learning algorithm then looks at learning properties that generalize across the different training tasks, and result in fast and efficient learning of the evaluation tasks.

One highly successful meta-learning algorithm has been *Model Agnostic Meta-Learning (MAML)* (Finn et al., 2017). At a high level, the MAML algorithm is comprised of two optimization loops. The outer loop (in the spirit of meta-learning) aims to find an effective *meta-initialization*, from which the inner loop can perform efficient *adaptation* – optimize parameters to solve new tasks with very few labelled examples. This algorithm, with deep neural networks as the underlying model, has been highly influential, with significant follow on work, such as first order variants (Nichol and Schulman, 2018), probabilistic extensions (Finn et al., 2018), augmentation with generative modelling (Rusu et al., 2018), and many others (Hsu et al., 2018; Finn and Levine, 2017; Grant et al., 2018; Triantafillou et al., 2019).

---

*Equal contribution

Despite the popularity of MAML, and the numerous followups and extensions, there remains a fundamental open question on the basic algorithm. Does the meta-initialization learned by the outer loop result in *rapid learning* on unseen test tasks (efficient but significant changes in the representations) or is the success primarily due to *feature reuse* (with the meta-initialization already providing high quality representations)? In this paper, we explore this question and its many surprising consequences. Our main contributions are:

- We perform layer freezing experiments and latent representational analysis of MAML, finding that feature reuse is the predominant reason for efficient learning.
- Based on these results, we propose the *ANIL (Almost No Inner Loop)* algorithm, a significant simplification to MAML that removes the inner loop updates for all but the head (final layer) of a neural network during training *and* inference. ANIL performs identically to MAML on standard benchmark few-shot classification and RL tasks and offers computational benefits over MAML.
- We study the effect of the head of the network, finding that once training is complete, the head can be removed, and the representations can be used without adaptation to perform unseen tasks, which we call the *No Inner Loop (NIL)* algorithm.
- We study different training regimes, e.g. multiclass classification, multitask learning, etc, and find that the task specificity of MAML/ANIL at training facilitate the learning of better features. We also find that multitask training, a popular baseline with no task specificity, performs worse than random features.
- We discuss rapid learning and feature reuse in the context of other meta-learning approaches.

## 2 RELATED WORK

MAML (Finn et al., 2017) is a highly popular meta-learning algorithm for few-shot learning, achieving competitive performance on several benchmark few-shot learning problems (Koch et al., 2015; Vinyals et al., 2016; Snell et al., 2017; Santoro et al., 2016; Ravi and Larochelle, 2016; Nichol and Schulman, 2018). It is part of the family of optimization-based meta-learning algorithms, with other members of this family presenting variations around how to learn the weights of the task-specific classifier. For example Lee and Choi (2018); Gordon et al. (2018); Bertinetto et al. (2018); Lee et al. (2019); Zhou et al. (2018) first learn functions to embed the support set and target examples of a few-shot learning task, before using the test support set to learn task specific weights to use on the embedded target examples. Harrison et al. (2018) also proceeds similarly, using a Bayesian approach. The method of Bao et al. (2019) explores a related approach, focusing on applications in text classification.

Of these optimization-based meta-learning algorithms, MAML has been especially influential, inspiring numerous direct extensions in recent literature (Antoniou et al., 2018; Finn et al., 2018; Grant et al., 2018; Rusu et al., 2018). Most of these extensions critically rely on the core structure of the MAML algorithm, incorporating an outer loop (for meta-training), and an inner loop (for task-specific adaptation), and there is little prior work analyzing why this central part of the MAML algorithm is practically successful. In this work, we focus on this foundational question, examining how and why MAML leads to effective few-shot learning. To do this, we utilize analytical tools such as Canonical Correlation Analysis (CCA) (Raghu et al., 2017; Morcos et al., 2018) and Centered Kernel Alignment (CKA) (Kornblith et al., 2019) to study the neural network representations learned with the MAML algorithm, which also demonstrates MAML's ability to learn effective features for few-shot learning.

Insights from this analysis lead to a simplified algorithm, ANIL, which almost completely removes the inner optimization loop with no reduction in performance. Prior works (Zintgraf et al., 2018; Javed and White, 2019) have proposed algorithms where some parameters are only updated in the outer loop and others only in the inner loop. However, these works are motivated by different questions, such as improving MAML's performance or learning better representations, rather than analysing rapid learning vs feature reuse in MAML.

## 3 MAML, RAPID LEARNING, AND FEATURE REUSE

Our goal is to understand whether the MAML algorithm efficiently solves new tasks due to *rapid learning* or *feature reuse*. In rapid learning, large representational and parameter changes occur during

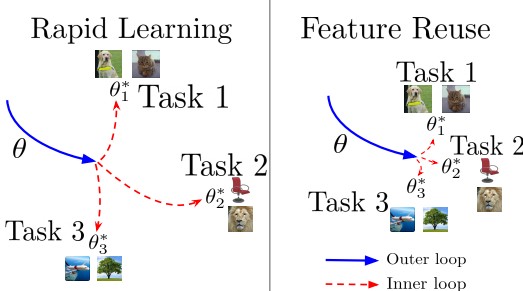

Figure 1: **Rapid learning and feature reuse paradigms.** In Rapid Learning, outer loop training leads to a parameter setting that is well-conditioned for fast learning, and inner loop updates result in significant task specialization. In Feature Reuse, the outer loop leads to parameter values corresponding to reusable features, from which the parameters do not move significantly in the inner loop.

adaptation to each new task as a result of favorable weight conditioning from the meta-initialization. In feature reuse, the meta-initialization already contains highly useful features that can mostly be reused as is for new tasks, so little task-specific adaptation occurs. Figure 1 shows a schematic of these two hypotheses.

We start off by overviewing the details of the MAML algorithm, and then we study the rapid learning vs feature reuse question via layer freezing experiments and analyzing latent representations of models trained with MAML. The results strongly support feature reuse as the predominant factor behind MAML's success. In Section 4, we explore the consequences of this, providing a significant simplification of MAML, the ANIL algorithm, and in Section 6, we outline the connections to meta-learning more broadly.

## 3.1 OVERVIEW OF MAML

The MAML algorithm finds an *initialization* for a neural network so that new tasks can be learnt with very few examples ($k$ examples from each class for $k$-shot learning) via two optimization loops:

- **Outer Loop:** Updates the initialization of the neural network parameters (often called the *meta-initialization*) to a setting that enables fast adaptation to new tasks.
- **Inner Loop:** Performs *adaptation*: takes the outer loop initialization, and, *separately for each task*, performs a few gradient updates over the $k$ labelled examples (the *support set*) provided for adaptation.

More formally, we first define our base model to be a neural network with meta-initialization parameters $\theta$; let this be represented by $f_\theta$. We have have a distribution $\mathcal{D}$ over tasks, and draw a batch $\{T_1, ..., T_B\}$ of $B$ tasks from $\mathcal{D}$. For each task $T_b$, we have a *support set* of examples $\mathcal{S}_{T_b}$, which are used for inner loop updates, and a *target set* of examples $\mathcal{Z}_{T_b}$, which are used for outer loop updates. Let $\theta_i^{(b)}$ signify $\theta$ after $i$ gradient updates for task $T_b$, and let $\theta_0^{(b)} = \theta$. In the inner loop, during each update, we compute

$$\theta_m^{(b)} = \theta_{m-1}^{(b)} - \alpha \nabla_{\theta_{m-1}^{(b)}} \mathcal{L}_{S_{T_b}}(f_{\theta_{m-1}^{(b)}(\theta)}) \tag{1}$$

for $m$ fixed across all tasks, where $\mathcal{L}_{S_{T_b}}(f_{\theta_{m-1}^{(b)}(\theta)})$ is the loss on the support set of $T_b$ after $m-1$ inner loop updates.

We then define the meta-loss as

$$\mathcal{L}_{meta}(\theta) = \sum_{b=1}^{B} \mathcal{L}_{\mathcal{Z}_{T_b}}(f_{\theta_m^{(b)}(\theta)})$$

where $\mathcal{L}_{Z_{T_b}}(f_{\theta_m^{(b)}(\theta)})$ is the loss on the target set of $T_b$ after $m$ inner loop updates, making clear the dependence of $f_{\theta_m^{(b)}}$ on $\theta$. The outer optimization loop then updates $\theta$ as

$$\theta = \theta - \eta \nabla_\theta \mathcal{L}_{meta}(\theta)$$

| Freeze layers | MiniImageNet-5way-1shot | MiniImageNet-5way-5shot |
|:---:|:---:|:---:|
| None | $46.9 \pm 0.2$ | $63.1 \pm 0.4$ |
| 1 | $46.5 \pm 0.3$ | $63.0 \pm 0.6$ |
| 1,2 | $46.4 \pm 0.4$ | $62.6 \pm 0.6$ |
| 1,2,3 | $46.3 \pm 0.4$ | $61.2 \pm 0.5$ |
| 1,2,3,4 | $46.3 \pm 0.4$ | $61.0 \pm 0.6$ |

Table 1: **Freezing successive layers (preventing inner loop adaptation) does not affect accuracy, supporting feature reuse.** To test the amount of feature reuse happening in the inner loop adaptation, we test the accuracy of the model when we freeze (prevent inner loop adaptation) a contiguous block of layers at test time. We find that freezing even all four convolutional layers of the network (all layers except the network head) hardly affects accuracy. This strongly supports the feature reuse hypothesis: layers don't have to change rapidly at adaptation time; they already contain good features from the meta-initialization.

At test time, we draw unseen tasks $\{T_1^{(test)}, ..., T_n^{(test)}\}$ from the task distribution, and evaluate the loss and accuracy on $\mathcal{Z}_{T_i^{(test)}}$ after inner loop adaptation using $\mathcal{S}_{T_i^{(test)}}$ (e.g. loss is $\mathcal{L}_{\mathcal{Z}_{T_i^{(test)}}}\left(f_{\theta_m^{(i)}(\theta)}\right)$).

## 3.2 Rapid Learning or Feature Reuse?

We now turn our attention to the key question: *Is MAML's efficacy predominantly due to rapid learning or feature reuse?* In investigating this question, there is an important distinction between the *head* (final layer) of the network and the earlier layers (the *body* of the network). In each few-shot learning task, there is a different alignment between the output neurons and classes. For instance, in task $\mathcal{T}_1$, the (wlog) five output neurons might correspond, in order, to the classes (dog, cat, frog, cupcake, phone), while for a different task, $\mathcal{T}_2$, they might correspond, in order, to (airplane, frog, boat, car, pumpkin). This means that the head must necessarily change for each task to learn the new alignment, and for the rapid learning vs feature reuse question, we are primarily interested in the behavior of the body of the network. We return to this in more detail in Section 5, where we present an algorithm (NIL) that does not use a head at test time.

To study rapid learning vs feature reuse in the network body, we perform two sets of experiments: (1) We evaluate few-shot learning performance when freezing parameters after MAML training, without test time inner loop adaptation; (2) We use representational similarity tools to directly analyze how much the network features and representations change through the inner loop. We use the MiniImageNet dataset, a popular standard benchmark for few-shot learning, and with the standard convolutional architecture in Finn et al. (2017). Results are averaged over three random seeds. Full implementation details are in Appendix B.

### 3.2.1 Freezing Layer Representations

To study the impact of the inner loop adaptation, we freeze a contiguous subset of layers of the network, during the inner loop at test time (after using the standard MAML algorithm, incorporating both optimization loops, for training). In particular, the frozen layers are not updated at all to the test time task, and must reuse the features learned by the meta-initialization that the outer loop converges to. We compare the few-shot learning accuracy when freezing to the accuracy when allowing inner loop adaptation.

Results are shown in Table 1. We observe that even when freezing all layers in the network body, performance hardly changes. This suggests that the meta-initialization has already learned good enough features that can be reused as is, without needing to perform any rapid learning for each test time task.

### 3.2.2 Representational Similarity Experiments

We next study how much the latent representations (the latent functions) learned by the neural network change during the inner loop adaptation phase. Following several recent works (Raghu et al., 2017; Saphra and Lopez, 2018; Morcos et al., 2018; Maheswaranathan et al., 2019; Raghu et al.,

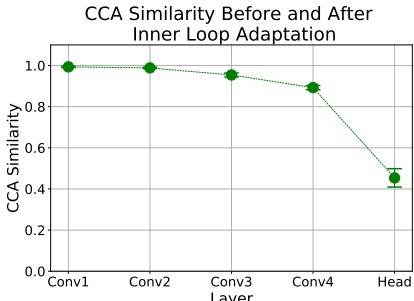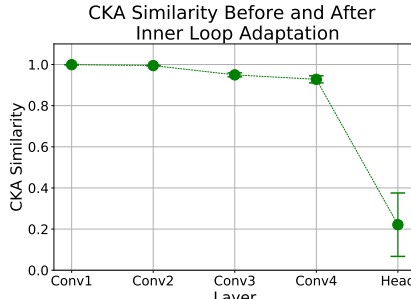

Figure 2: **High CCA/CKA similarity between representations before and after adaptation for all layers except the head.** We compute CCA/CKA similarity between the representation of a layer before the inner loop adaptation and after adaptation. We observe that for all layers except the head, the CCA/CKA similarity is almost 1, indicating perfect similarity. This suggests that these layers do not change much during adaptation, but mostly perform feature reuse. Note that there is a slight dip in similarity in the higher conv layers (e.g. conv3, conv4); this is likely because the slight representational differences in conv1, conv2 have a compounding effect on the representations of conv3, conv4. The head of the network must change significantly during adaptation, and this is reflected in the much lower CCA/CKA similarity.

2019; Gotmare et al., 2018; Bau et al., 2018) we measure this by applying Canonical Correlation Analysis (CCA) to the latent representations of the network. CCA provides a way to the compare representations of two (latent) layers $L_1, L_2$ of a neural network, outputting a similarity score between $0$ (not similar at all) and $1$ (identical). For full details, see Raghu et al. (2017); Morcos et al. (2018). In our analysis, we take $L_1$ to be a layer before the inner loop adaptation steps, and $L_2$ after the inner loop adaptation steps. We compute CCA similarity between $L_1, L_2$, averaging the similarity score across different random seeds of the model and different test time tasks. Full details are in Appendix B.2

The result is shown in Figure 2, left pane. Representations in the body of the network (the convolutional layers) are highly similar, with CCA similarity scores of $> 0.9$, indicating that the inner loop induces little to no functional change. By contrast, the head of the network, which does change significantly in the inner loop, has a CCA similarity of less than $0.5$. To further validate this, we also compute CKA (Centered Kernel Alignment) (Kornblith et al., 2019) (Figure 2 right), another similarity metric for neural network representations, which illustrates the same pattern. These representational analysis results strongly support the feature reuse hypothesis, with further results in the Appendix, Sections B.3 and B.4 providing yet more evidence.

### 3.2.3 FEATURE REUSE HAPPENS EARLY IN LEARNING

Having observed that the inner loop does not significantly affect the learned representations with a fully trained model, we extend our analysis to see whether the inner loop affects representations and features earlier on in training. We take MAML models at 10000, 20000, and 30000 iterations into training, perform freezing experiments (as in Section 3.2.1) and representational similarity experiments (as in Section 3.2.2).

Results in Figure 3 show the same patterns from early in training, with CCA similarity between activations pre and post inner loop update on MiniImageNet-5way-5shot being very high for the body (just like Figure 2), and similar to Table 1, test accuracy remaining approximately the same when freezing contiguous subsets of layers, even when freezing all layers of the network body. This shows that even early on in training, significant feature reuse is taking place, with the inner loop having minimal effect on learned representations and features. Results for 1shot MiniImageNet are in Appendix B.5, and show very similar trends.

## 4 THE ANIL (ALMOST NO INNER LOOP) ALGORITHM

In the previous section we saw that for all layers except the head of the neural network, the meta-initialization learned by the outer loop of MAML results in very good features that can be reused

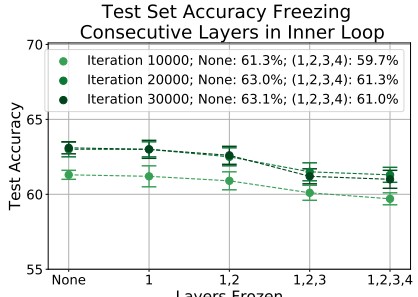 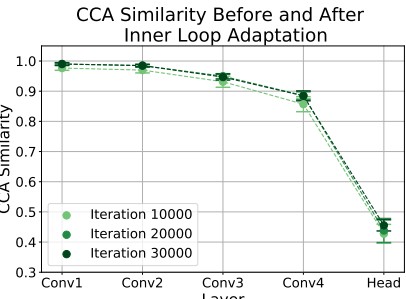

Figure 3: **Inner loop updates have little effect on learned representations from early on in learning.** Left pane: we freeze contiguous blocks of layers (no adaptation at test time), on MiniImageNet-5way-5shot and see almost identical performance. Right pane: representations of all layers except the head are highly similar pre/post adaptation – i.e. features are being reused. This is true from early (iteration 10000) in training.

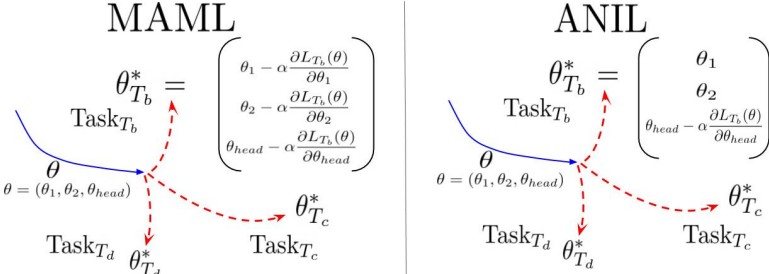

Figure 4: **Schematic of MAML and ANIL algorithms.** The difference between the MAML and ANIL algorithms: in MAML (left), the inner loop (task-specific) gradient updates are applied to all parameters $\theta$, which are initialized with the meta-initialization from the outer loop. In ANIL (right), only the parameters corresponding to the network head $\theta_{head}$ are updated by the inner loop, during training **and** testing.

as is on new tasks. Inner loop adaptation does not significantly change the representations of these layers, even from early on in training. This suggests a natural simplification of the MAML algorithm: the *ANIL (Almost No Inner Loop) algorithm.*

In ANIL, during training **and** testing, we *remove* the inner loop updates for the network body, and apply inner loop adaptation *only to the head.* The head requires the inner loop to allow it to align to the different classes in each task. In Section 5.1 we consider another variant, the *NIL (No Inner Loop) algorithm*, that removes the head entirely at test time, and uses learned features and cosine similarity to perform effective classification, thus avoiding inner loop updates altogether.

For the ANIL algorithm, mathematically, let $\theta = (\theta_1, ..., \theta_l)$ be the (meta-initialization) parameters for the $l$ layers of the network. Following the notation of Section 3.1, let $\theta_m^{(b)}$ be the parameters after $m$ inner gradient updates for task $\mathcal{T}_b$. In ANIL, we have that:

$$\theta_m^{(b)} = \left( \theta_1, \dots, (\theta_l)_{m-1}^{(b)} - \alpha \nabla_{(\theta_l)_{m-1}^{(b)}} \mathcal{L}_{S_b} (f_{\theta_{m-1}^{(b)}}) \right)$$

i.e. only the final layer gets the inner loop updates. As before, we then define the meta-loss, and compute the outer loop gradient update. The intuition for ANIL arises from Figure 3, where we observe that inner loop updates have little effect on the network body even early in training, suggesting the possibility of removing them entirely. Note that this is distinct to the freezing experiments, where we only removed the inner loop at inference time. Figure 4 presents the difference between MAML and ANIL, and Appendix C.1 considers a simple example of the gradient update in ANIL, showing how the ANIL update differs from MAML.

**Computational benefit of ANIL:** As ANIL almost has no inner loop, it significantly speeds up both training and inference. We found an average speedup of 1.7x per training iteration over MAML and an average speedup of 4.1x per inference iteration. In Appendix C.5 we provide the full results.

| Method | Omniglot-20way-1shot | Omniglot-20way-5shot | MiniImageNet-5way-1shot | MiniImageNet-5way-5shot |
|--------|----------------------|----------------------|-------------------------|-------------------------|
| MAML   | $93.7 \pm 0.7$       | $96.4 \pm 0.1$       | $46.9 \pm 0.2$          | $63.1 \pm 0.4$          |
| ANIL   | $96.2 \pm 0.5$       | $98.0 \pm 0.3$       | $46.7 \pm 0.4$          | $61.5 \pm 0.5$          |

| Method | HalfCheetah-Direction | HalfCheetah-Velocity | 2D-Navigation |
|--------|-----------------------|----------------------|---------------|
| MAML   | $170.4 \pm 21.0$      | $-139.0 \pm 18.9$    | $-20.3 \pm 3.2$ |
| ANIL   | $363.2 \pm 14.8$      | $-120.9 \pm 6.3$     | $-20.1 \pm 2.3$ |

Table 2: **ANIL matches the performance of MAML on few-shot image classification and RL.** On benchmark few-shot classification tasks MAML and ANIL have comparable accuracy, and also comparable average return (the higher the better) on standard RL tasks (Finn et al., 2017).

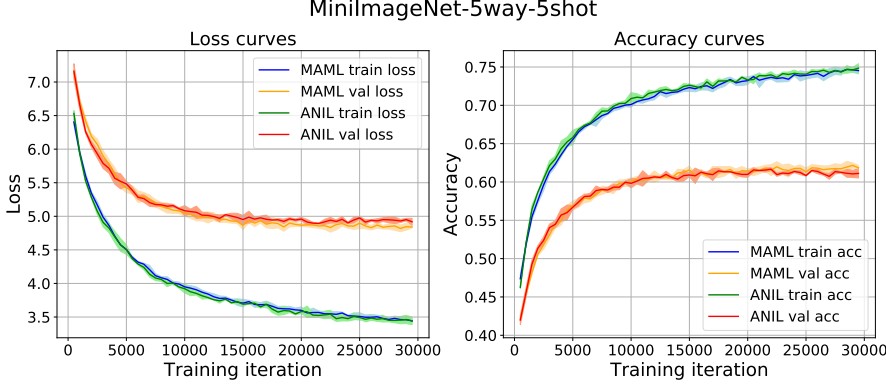

Figure 5: **MAML and ANIL learn very similarly**. Loss and accuracy curves for MAML and ANIL on MiniImageNet-5way-5shot, illustrating how MAML and ANIL behave similarly through the training process.

**Results of ANIL on Standard Benchmarks:** We evaluate ANIL on few-shot image classification and RL benchmarks, using the same model architectures as the original MAML authors, for both supervised learning and RL. Further implementation details are in Appendix C.4. The results in Table 2 (mean and standard deviation of performance over three random initializations) show that ANIL matches the performance of MAML on both few-shot classification (accuracy) and RL (average return, the higher the better), demonstrating that the inner loop adaptation of the body is unnecessary for learning good features.

**MAML and ANIL Models Show Similar Behavior:** MAML and ANIL perform equally well on few-shot learning benchmarks, illustrating that removing the inner loop during training does not hinder performance. To study the behavior of MAML and ANIL models further, we plot learning curves for both algorithms on MiniImageNet-5way-5shot, Figure 5. We see that loss and accuracy for both algorithms look very similar throughout training. We also look at CCA and CKA scores of the representations learned by both algorithms, Table 3. We observe that MAML-ANIL representations have the same average similarity scores as MAML-MAML and ANIL-ANIL representations, suggesting both algorithms learn comparable features (removing the inner loop doesn't change the kinds of features learned.) Further learning curves and representational similarity results are presented in Appendices C.2 and C.3.

## 5 CONTRIBUTIONS OF THE NETWORK HEAD AND BODY

So far, we have seen that MAML predominantly relies on feature reuse, with the network body (all layers except the last layer) already containing good features at meta-initialization. We also observe that such features can be learned even without inner loop adaptation during training (ANIL algorithm). The head, however, requires inner loop adaptation to enable task specificity.

| Model Pair | CCA Similarity | CKA Similarity |
|---|---|---|
| MAML-MAML | 0.51 | 0.83 |
| ANIL-ANIL | 0.51 | 0.86 |
| ANIL-MAML | 0.50 | 0.83 |

Table 3: **MAML and ANIL models learn comparable representations.** Comparing CCA/CKA similarity scores of the of MAML-ANIL representations (averaged over network body), and MAML-MAML and ANIL-ANIL similarity scores (across different random seeds) shows algorithmic differences between MAML/ANIL does not result in vastly different types of features learned.

| Method | Omniglot-20way-1shot | Omniglot-20way-5shot | MiniImageNet-5way-1shot | MiniImageNet-5way-5shot |
|---|---|---|---|---|
| MAML | $93.7 \pm 0.7$ | $96.4 \pm 0.1$ | $46.9 \pm 0.2$ | $63.1 \pm 0.4$ |
| ANIL | $96.2 \pm 0.5$ | $98.0 \pm 0.3$ | $46.7 \pm 0.4$ | $61.5 \pm 0.5$ |
| NIL | $96.7 \pm 0.3$ | $98.0 \pm 0.04$ | $48.0 \pm 0.7$ | $62.2 \pm 0.5$ |

Table 4: **NIL algorithm performs as well as MAML and ANIL on few-shot image classification.** Performance of MAML, ANIL, and NIL on few-shot image classification benchmarks. We see that with no test-time inner loop, and just learned features, NIL performs comparably to MAML and ANIL, indicating the strength of the learned features, and the relative lack of importance of the head at test time.

In this section, we explore the contributions of the network head and body. We first ask: *How important is the head at test time, when good features have already been learned?* Motivating this question is that the features in the body of the network needed no adaptation at inference time, so perhaps they are themselves sufficient to perform classification, with no head. In Section 5.1, we find that test time performance is entirely determined by the quality of these representations, and we can use similarity of the frozen meta-initialization representations to perform unseen tasks, *removing the head entirely*. We call this the NIL (No Inner Loop) algorithm.

Given this result, we next study how useful the head is at training (in ensuring the network body learns good features). We look at multiple different training regimes (some without the head) for the network body, and evaluate the quality of the representations. We find that MAML/ANIL result in the best representations, demonstrating the importance of the head during training for feature learning.

## 5.1 THE HEAD AT TEST TIME AND THE NIL (NO INNER LOOP) ALGORITHM

We study how important the head and task specific alignment are when good features have *already* been learned (through training) by the meta-initialization. At test time, we find that the representations can be used directly, with *no* adaptation, which leads to the *No Inner Loop (NIL) algorithm*:

1. Train a few-shot learning model with ANIL/MAML algorithm as standard. We use ANIL training.

2. At test time, remove the head of the trained model. For each task, first pass the $k$ labelled examples (support set) through the body of the network, to get their penultimate layer representations. Then, for a test example, compute cosine similarities between its penultimate layer representation and those of the support set, using these similarities to weight the support set labels, as in Vinyals et al. (2016).

The results for the NIL algorithm, following ANIL training, on few-shot classification benchmarks are given in Table 4. Despite having no network head and no task specific adaptation, NIL performs comparably to MAML and ANIL. This demonstrates that the features learned by the network body when training with MAML/ANIL (and reused at test time) are the critical component in tackling these benchmarks.

## 5.2 TRAINING REGIMES FOR THE NETWORK BODY

The NIL algorithm and results of Section 5.1 lead to the question of how important task alignment and the head are during training to ensure good features. Here, we study this question by examining the quality of features arising from different training regimes for the body. We look at (i) MAML

| Method | MiniImageNet-5way-1shot | MiniImageNet-5way-5shot |
|---|---|---|
| MAML training-NIL head | $48.4 \pm 0.3$ | $61.5 \pm 0.8$ |
| ANIL training-NIL head | $48.0 \pm 0.7$ | $62.2 \pm 0.5$ |
| Multiclass training-NIL head | $39.7 \pm 0.3$ | $54.4 \pm 0.5$ |
| Multitask training-NIL head | $26.5 \pm 1.1$ | $34.2 \pm 3.5$ |
| Random features-NIL head | $32.9 \pm 0.6$ | $43.2 \pm 0.5$ |
| NIL training-NIL head | $38.3 \pm 0.6$ | $43.0 \pm 0.2$ |

Table 5: **MAML/ANIL training leads to superior features learned, supporting importance of head at training.** Training with MAML/ANIL leads to superior performance over other methods which do not have task specific heads, supporting the importance of the head at training.

and ANIL training; (ii) *multiclass classification*, where all of the training data and classes (from which training tasks are drawn) are used to perform standard classification; (iii) *multitask training*, a standard baseline, where no inner loop or task specific head is used, but the network is trained on all the tasks at the same time; (iv) *random features*, where the network is not trained at all, and features are frozen after random initialization; (v) NIL at training time, where there is no head and cosine distance on the representations is used to get the label.

After training, we apply the NIL algorithm to evaluate test performance, and quality of features learned at training. The results are shown in Table 5. MAML and ANIL training performs best. Multitask training, which has no task specific head, performs the worst, even worse than random features (adding evidence for the need for task specificity at training to facilitate feature learning.) Using NIL during training performs worse than MAML/ANIL. These results suggest that the head is important at training to learn good features in the network body.

In Appendix D.1, we study test time performance variations from using a MAML/ANIL head instead of NIL, finding (as suggested by Section 5.1) very little performance difference. Additional results on similarity between the representations of different training regimes is given in Appendix D.2.

## 6 FEATURE REUSE IN OTHER META-LEARNING ALGORITHMS

Up till now, we have closely examined the MAML algorithm, and have demonstrated empirically that the algorithm's success is primarily due to feature reuse, rather than rapid learning. We now discuss rapid learning vs feature reuse more broadly in meta-learning. By combining our results with an analysis of evidence reported in prior work, we find support for many meta-learning algorithms succeeding via feature reuse, identifying a common theme characterizing the operating regime of much of current meta-learning.

### 6.1 OPTIMIZATION AND MODEL BASED META-LEARNING

MAML falls within the broader class of *optimization based* meta-learning algorithms, which at inference time, directly optimize model parameters for a new task using the support set. MAML has inspired many other optimization-based algorithms, which utilize the same two-loop structure (Lee and Choi, 2018; Rusu et al., 2018; Finn et al., 2018). Our analysis so far has thus yielded insights into the feature reuse vs rapid learning question for this class of algorithms. Another broad class of meta-learning consists of *model based* algorithms, which also have notions of rapid learning and feature reuse.

In the model-based setting, the meta-learning model's parameters are *not* directly optimized for the specific task on the support set. Instead, the model typically conditions its output on some representation of the task definition. One way to achieve this conditioning is to *jointly encode* the entire support set in the model's latent representation (Vinyals et al., 2016; Sung et al., 2018), enabling it to adapt to the characteristics of each task. This constitutes rapid learning for model based meta-learning algorithms.

An alternative to joint encoding would be to encode each member of the support set independently, and apply a cosine similarity rule (as in Vinyals et al. (2016)) to classify an unlabelled example. This mode of operation is purely feature reuse – we do not use information defining the task to directly influence the decision function.

If joint encoding gave significant test-time improvement over non-joint encoding, then this would suggest that rapid learning of the test-time task is taking place, as task specific information is being utilized to influence the model's decision function. However, on analyzing results in prior literature, this improvement appears to be minimal. Indeed, in e.g. Matching Networks (Vinyals et al., 2016), using joint encoding one reaches 44.2% accuracy on MiniImageNet-5way-1shot, whereas with independent encoding one obtains 41.2%: a small difference. More refined models suggest the gap is even smaller. For instance, in Chen et al. (2019), many methods for one shot learning were re-implemented and studied, and baselines without joint encoding achieved 48.24% accuracy in MiniImageNet-5way-1shot, whilst other models using joint encoding such as Relation Net (Sung et al., 2018) achieve very similar accuracy of 49.31% (they also report MAML, at 46.47%). As a result, we believe that the dominant mode of "feature reuse" rather than "rapid learning" is what has currently dominated both MAML-styled optimization based meta-learning *and* model based meta-learning.

## 7  CONCLUSION

In this paper, we studied a fundamental question on whether the highly successful MAML algorithm relies on rapid learning or feature reuse. Through a series of experiments, we found that *feature reuse* is the dominant component in MAML's efficacy on benchmark datasets. This insight led to the ANIL (Almost No Inner Loop) algorithm, a simplification of MAML that has identical performance on standard image classification and reinforcement learning benchmarks, and provides computational benefits. We further study the importance of the head (final layer) of a neural network trained with MAML, discovering that the body (lower layers) of a network is sufficient for few-shot classification at test time, allowing us to remove the network head for testing (NIL algorithm) and still match performance. We connected our results to the broader literature in meta-learning, identifying feature reuse to be a common mode of operation for other meta-learning algorithms also. Based off of our conclusions, future work could look at developing and analyzing new meta-learning algorithms that perform more rapid learning, which may expand the datasets and problems amenable to these techniques. We note that our study mainly considered benchmark datasets, such as Omniglot and MiniImageNet. It is an interesting future direction to consider rapid learning and feature reuse in MAML on other few-shot learning datasets, such as those from Triantafillou et al. (2019).

## ACKNOWLEDGEMENTS

The authors thank Geoffrey Hinton, Chelsea Finn, Hugo Larochelle and Chiyuan Zhang for helpful feedback on the methods and results.

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

## A  FEW-SHOT IMAGE CLASSIFICATION DATASETS AND EXPERIMENTAL SETUPS

We consider the few-shot learning paradigm for image classification to evaluate MAML and ANIL. We evaluate using two datasets often used for few-shot multiclass classification – the Omniglot dataset and the MiniImageNet dataset.

**Omniglot:**  The Omniglot dataset consists of over 1600 different handwritten character classes from 23 alphabets. The dataset is split on a character-level, so that certain characters are in the training set, and others in the validation set. We consider the 20-way 1-shot and 20-way 5-shot tasks on this dataset, where at test time, we wish our classifier to discriminate between 20 randomly chosen character classes from the held-out set, given only 1 or 5 labelled example(s) from each class from this set of 20 testing classes respectively. The model architecture used is identical to that in the original MAML paper, namely: 4 modules with a 3 x 3 convolutions and 64 filters with a stride of 2, followed by batch normalization, and a ReLU nonlinearity. The Omniglot images are downsampled to 28 x 28, so the dimensionality of the last hidden layer is 64. The last layer is fed into a 20-way softmax. Our models are trained using a batch size of 16, 5 inner loop updates, and an inner learning rate of 0.1.

**MiniImageNet:**  The MiniImagenet dataset was proposed by Ravi and Larochelle (2016), and consists of 64 training classes, 12 validation classes, and 24 test classes. We consider the 5-way 1-shot and 5-way 5-shot tasks on this dataset, where the test-time task is to classify among 5 different randomly chosen validation classes, given only 1 and 5 labelled examples respectively. The model architecture is again identical to that in the original paper: 4 modules with a 3 x 3 convolutions and 32 filters, followed by batch normalization, ReLU nonlinearity, and 2 x 2 max pooling. Our models are trained using a batch size of 4. 5 inner loop update steps, and an inner learning rate of 0.01 are used. 10 inner gradient steps are used for evaluation at test time.

## B  ADDITIONAL DETAILS AND RESULTS: FREEZING AND REPRESENTATIONAL SIMILARITY

In this section, we provide further experimental details and results from freezing and representational similarity experiments.

### B.1  EXPERIMENTAL DETAILS

We concentrate on MiniImageNet for our experiments in Section 3.2, as it is more complex than Omniglot.

The model architecture used for our experiments is identical to that in the original paper: 4 modules with a $3 \times 3$ convolutions and 32 filters, followed by batch normalization, ReLU nonlinearity, and $2 \times 2$ max pooling. Our models are trained using a batch size of 4, 5 inner loop update steps, and an inner learning rate of 0.01. 10 inner gradient steps are used for evaluation at test time. We train models 3 times with different random seeds. Models were trained for 30000 iterations.

### B.2  DETAILS OF REPRESENTATIONAL SIMILARITY

CCA takes in as inputs $L_1 = \{z_1^{(1)}, z_2^{(1)}, ..., z_m^{(1)}\}$ and $L_2 = \{z_1^{(2)}, z_2^{(1)}, ..., z_n^{(2)}\}$, where $L_1, L_2$ are layers, and $z_i^{(j)}$ is a neuron activation vector: the vector of outputs of neuron $i$ (of layer $L_j$) over a set of inputs $X$. It then finds linear combinations of the neurons in $L_1$ and neurons in $L_2$ so that the resulting activation vectors are maximally correlated, which is summarized in the canonical correlation coefficient. Iteratively repeating this process gives a similarity score (in $[0, 1]$ with 1 identical and 0 completely different) between the representations of $L_1$ and $L_2$.

We apply this to compare corresponding layers of two networks, net1 and net2, where net1 and net2 might differ due to training step, training method (ANIL vs MAML) or the random seed. When comparing convolutional layers, as described in Raghu, we perform the comparison over channels,

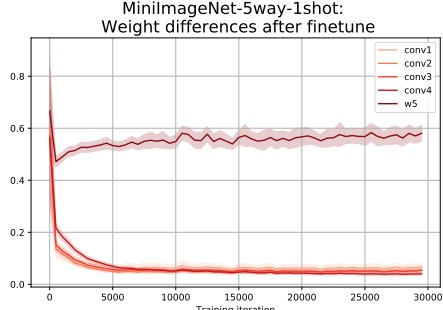 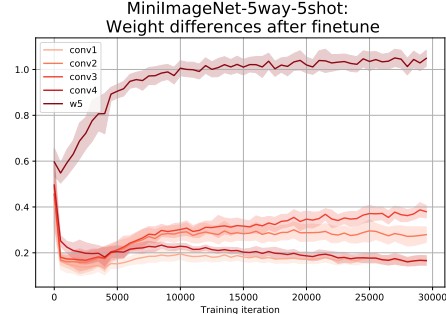

Figure 6: **Euclidean distance before and after finetuning for MiniImageNet**. We compute the average (across tasks) Euclidean distance between the weights before and after inner loop adaptation, separately for different layers. We observe that all layers except for the final layer show very little difference before and after inner loop adaptation, suggesting significant feature reuse.

flattening out over all of the spatial dimensions, and then taking the mean CCA coefficient. We average over three random repeats.

### B.3 SIMILARITY BEFORE AND AFTER INNER LOOP WITH EUCLIDEAN DISTANCE

In addition to assessing representational similarity with CCA/CKA, we also consider the simpler measure of Euclidean distance, capturing how much weights of the network change during the inner loop update (task-specific finetuning). We note that this experiment does not assess functional changes on inner loop updates as well as the CCA experiments do; however, they serve to provide useful intuition.

We plot the per-layer average Euclidean distance between the initialization $\theta$ and the finetuned weights $\theta_m^{(b)}$ across different tasks $T_b$, i.e.

$$\frac{1}{N} \sum_{b=1}^{N} ||(\theta_l) - (\theta_l)_m^{(b)}||$$

across different layers $l$, for MiniImageNet in Figure 6. We observe that very quickly after the start of training, all layers except for the last layer have small Euclidean distance difference before and after finetuning, suggesting significant feature reuse. (Note that this is despite the fact that these layers have more parameters than the final layer.)

### B.4 CCA SIMILARITY ACROSS RANDOM SEEDS

The experiment in Section 3.2.2 compared representational similarity of $L_1$ and $L_2$ at different points in training (before/after inner loop adaptation) but corresponding to the same random seed. To complete the picture, it is useful to study whether representational similarity across *different* random seeds is also mostly unaffected by the inner loop adaptation. This motivates four natural comparisons: assume layer $L_1$ is from the first seed, and layer $L_2$ is from the second seed. Then we can compute the representational similarity between ($L_1$ pre, $L_2$ pre), ($L_1$ pre, $L_2$ post), ($L_1$ post, $L_2$ pre) and ($L_1$ post, $L_2$ post), where pre/post signify whether we take the representation before or after adaptation.

Prior work has shown that neural network representations may vary across different random seeds (Raghu et al., 2017; Morcos et al., 2018; Li et al., 2015; Wang et al., 2018), organically resulting in CCA similarity scores much less than 1. So to identify the effect of the inner loop on the representation, we plot the CCA similarities of (i) ($L_1$ pre, $L_2$ pre) against ($L_1$ pre, $L_2$ post) and (ii) ($L_1$ pre, $L_2$ pre) against ($L_1$ post, $L_2$ pre) and (iii) ($L_1$ pre, $L_2$ pre) against ($L_1$ post, $L_2$ post) separately across the different random seeds and different layers. We then compute the line of best fit for each plot. If the line of best fit fits the data and is close to $y = x$, this suggests that the inner loop adaptation doesn't affect the features much – the similarity before adaptation is very close to the similarity after adaptation.

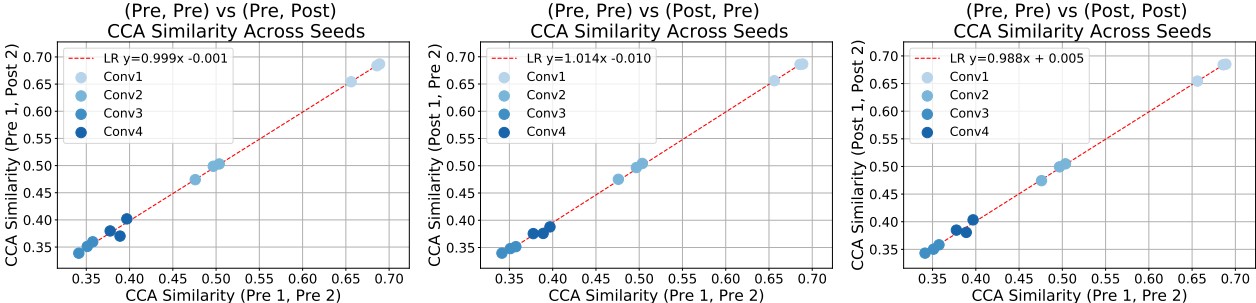

Figure 7: **Computing CCA similarity pre/post adaptation across different random seeds further demonstrates that the inner loop doesn't change representations significantly.** We compute CCA similarity of $L_1$ from seed 1 and $L_2$ from seed 2, varying whether we take the representation *pre* (before) adaptation or *post* (after) adaptation. To isolate the effect of adaptation from inherent variation in the network representation across seeds, we plot CCA similarity of of the representations before adaptation against representations after adaptation in three different combinations: (i) ($L_1$ pre, $L_2$ pre) against ($L_1$ pre, $L_1$ post), (ii) ($L_1$ pre, $L_2$ pre) against ($L_1$ pre, $L_1$ post) (iii) ($L_1$ pre, $L_2$ pre) against ($L_1$ post, $L_2$ post). We do this separately across different random seeds and different layers. Then, we compute a line of best fit, finding that in all three plots, it is almost identical to $y = x$, demonstrating that the representation does not change significantly pre/post adaptation. Furthermore a computation of the coefficient of determination $R^2$ gives $R^2 \approx 1$, illustrating that the data is well explained by this relation. In Figure 8, we perform this comparison with CKA, observing the same high level conclusions.

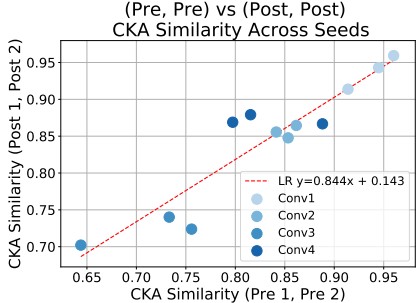

Figure 8: We perform the same comparison as in Figure 7, but with CKA instead. There is more variation in the similarity scores, but we still see a strong correlation between (Pre, Pre) and (Post, Post) comparisons, showing that representations do not change significantly over the inner loop.

The results are shown in Figure 7. In all of the plots, we see that the line of best fit is almost exactly $y = x$ (even for the pre/pre vs post/post plot, which could conceivably be more different as both seeds change) and a computation of the coefficient of determination $R^2$ gives $R^2 \approx 1$ for all three plots. Putting this together with Figure 2, we can conclude that the inner loop adaptation step doesn't affect the representation learned by any layer except the head, and that the learned representations and features are mostly reused as is for the different tasks.

### B.5    MINIIMAGENET-5WAY-1SHOT FREEZING AND CCA OVER TRAINING

Figure 9 shows that from early on in training, on MiniImageNet-5way-1shot, that the CCA similarity between activations pre and post inner loop update is very high for all layers but the head. We further see that the validation set accuracy suffers almost no decrease if we remove the inner loop updates and freeze all layers but the head. This shows that even early on in training, the inner loop appears to have minimal effect on learned representations and features. This supplements the results seen in Figure 3 on MiniImageNet-5way-5shot.

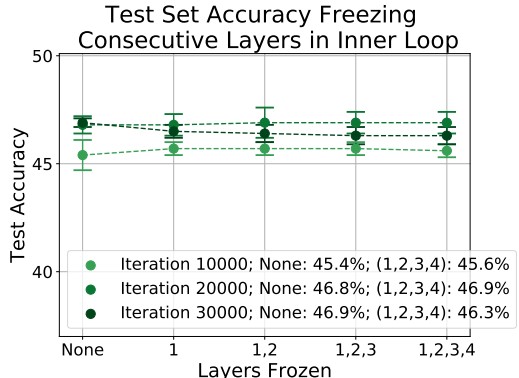 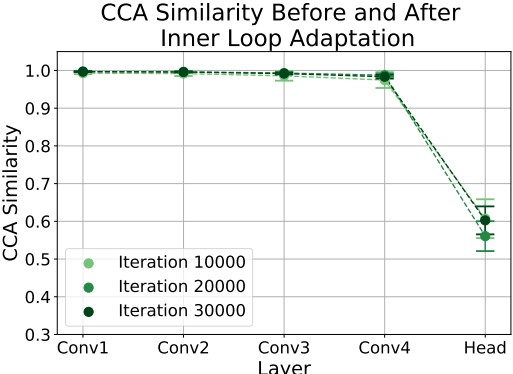

Figure 9: **Inner loop updates have little effect on learned representations from early on in learning.** We consider freezing and representational similarity experiments for MiniImageNet-5way-1shot. We see that early on in training (from as few as 10k iterations in), the inner loop updates have little effect on the learned representations and features, and that removing the inner loop updates for all layers but the head have little-to-no impact on the validation set accuracy.

## C ANIL ALGORITHM: MORE DETAILS

In this section, we provide more details about the ANIL algorithm, including an example of the ANIL update, implementation details, and further experimental results.

### C.1 AN EXAMPLE OF THE ANIL UPDATE

Consider a simple, two layer linear network with a single hidden unit in each layer: $\hat{y}(x; \boldsymbol{\theta}) = \theta_2(\theta_1 x)$. In this example, $\theta_2$ is the *head*. Consider the 1-shot regression problem, where we have access to examples $\left\{ (x_1^{(t)}, y_1^{(t)}), (x_2^{(t)}, y_2^{(t)}) \right\}$ for tasks $t = 1, \ldots, T$. Note that $(x_1^{(t)}, y_1^{(t)})$ is the (example, label) pair in the meta-training set (used for inner loop adaptation – support set), and $(x_2^{(t)}, y_2^{(t)})$ is the pair in the meta-validation set (used for the outer loop update – target set).

In the few-shot learning setting, we firstly draw a set of $N$ tasks and labelled examples from our meta-training set: $\left\{ (x_1^{(1)}, y_1^{(1)}), \ldots, (x_1^{(N)}, y_1^{(N)}) \right\}$. Assume for simplicity that we only apply one gradient step in the inner loop. The inner loop updates for each task are thus defined as follows:

$$\theta_1^{(t)} \leftarrow \theta_1 - \frac{\partial L(\hat{y}(x_1^{(t)}; \boldsymbol{\theta}), y_1^{(t)})}{\partial \theta_1} \tag{1}$$

$$\theta_2^{(t)} \leftarrow \theta_2 - \frac{\partial L(\hat{y}(x_1^{(t)}; \boldsymbol{\theta}), y_1^{(t)})}{\partial \theta_2} \tag{2}$$

where $L(\cdot, \cdot)$ is the loss function, (e.g. mean squared error) and $\theta_i^{(t)}$ refers to a parameter after inner loop update for task $t$.

The task-adapted parameters for MAML and ANIL are as follows. Note how only the head parameters change per-task in ANIL:

$$\boldsymbol{\theta}_{\text{MAML}}^{(t)} = \left[ \theta_1^{(t)}, \theta_2^{(t)} \right] \tag{3}$$

$$\boldsymbol{\theta}_{\text{ANIL}}^{(t)} = \left[ \theta_1, \theta_2^{(t)} \right] \tag{4}$$

In the outer loop update, we then perform the following operations using the data from the meta-validation set:

$$\theta_1 \leftarrow \theta_1 - \sum_{t=1}^{N} \frac{\partial L(\hat{y}(x_2^{(t)}; \boldsymbol{\theta}^{(t)}), y_2^{(t)})}{\partial \theta_1} \tag{5}$$

$$\theta_2 \leftarrow \theta_2 - \sum_{t=1}^{N} \frac{\partial L(\hat{y}(x_2^{(t)}; \boldsymbol{\theta}^{(t)}), y_2^{(t)})}{\partial \theta_2} \tag{6}$$

Considering the update for $\theta_1$ in more detail for our simple, two layer, linear network (the case for $\theta_2$ is analogous), we have the following update for MAML:

$$\theta_1 \leftarrow \theta_1 - \sum_{t=1}^{N} \frac{\partial L(\hat{y}(x_2^{(t)}; \boldsymbol{\theta}_{\text{MAML}}^{(t)}), y_2^{(t)})}{\partial \theta_1} \tag{7}$$

$$\hat{y}(x_2^{(t)}; \boldsymbol{\theta}_{\text{MAML}}^{(t)}) = \left( \left[ \theta_2 - \frac{\partial L(\hat{y}(x_1^{(t)}; \boldsymbol{\theta}), y_1^{(t)})}{\partial \theta_2} \right] \cdot \left[ \theta_1 - \frac{\partial L(\hat{y}(x_1^{(t)}; \boldsymbol{\theta}), y_1^{(t)})}{\partial \theta_1} \right] \cdot x_2 \right) \tag{8}$$

For ANIL, on the other hand, the update will be:

$$\theta_1 \leftarrow \theta_1 - \sum_{t=1}^{N} \frac{\partial L(\hat{y}(x_2^{(t)}; \boldsymbol{\theta}_{\text{ANIL}}^{(t)}), y_2^{(t)})}{\partial \theta_1} \tag{9}$$

$$\hat{y}(x_2^{(t)}; \boldsymbol{\theta}_{\text{ANIL}}^{(t)}) = \left( \left[ \theta_2 - \frac{\partial L(\hat{y}(x_1^{(t)}; \boldsymbol{\theta}), y_1^{(t)})}{\partial \theta_2} \right] \cdot \theta_1 \cdot x_2 \right) \tag{10}$$

Note the lack of inner loop update for $\theta_1$, and how we do not remove second order terms in ANIL (unlike in first-order MAML); second order terms still persist through the derivative of the inner loop update for the head parameters.

## C.2 ANIL LEARNS ALMOST IDENTICALLY TO MAML

We implement ANIL on MiniImageNet and Omniglot, and generate learning curves for both algorithms in Figure 10. We find that learning proceeds almost identically for ANIL and MAML, showing that removing the inner loop has little effect on the learning dynamics.

## C.3 ANIL AND MAML LEARN SIMILAR REPRESENTATIONS

We compute CCA similarities across representations in a MAML seed and an ANIL seed, and then plot these against the same MAML seed representation compared to a different MAML seed (and similarly for ANIL). We find a strong correlation between these similarities (Figure 11), which suggests that MAML and ANIL are learning similar representations, despite their algorithmic differences. (ANIL and MAML are about as similar to each other as two ANILs are to each other, or two MAMLs are to each other.)

## C.4 ANIL IMPLEMENTATION DETAILS

**Supervised Learning Implementation:** We used the TensorFlow MAML implementation open-sourced by the original authors (Finn et al., 2017). We used the same model architectures as in the original MAML paper for our experiments, and train models 3 times with different random seeds. All models were trained for 30000 iterations, with a batch size of 4, 5 inner loop update steps, and an inner learning rate of 0.01. 10 inner gradient steps were used for evaluation at test time.

**Reinforcement Learning Implementation:** We used the open source PyTorch implementation of MAML for RL [1], due to challenges encountered when running the open-sourced TensorFlow

---

[1]https://github.com/tristandeleu/pytorch-maml-rl

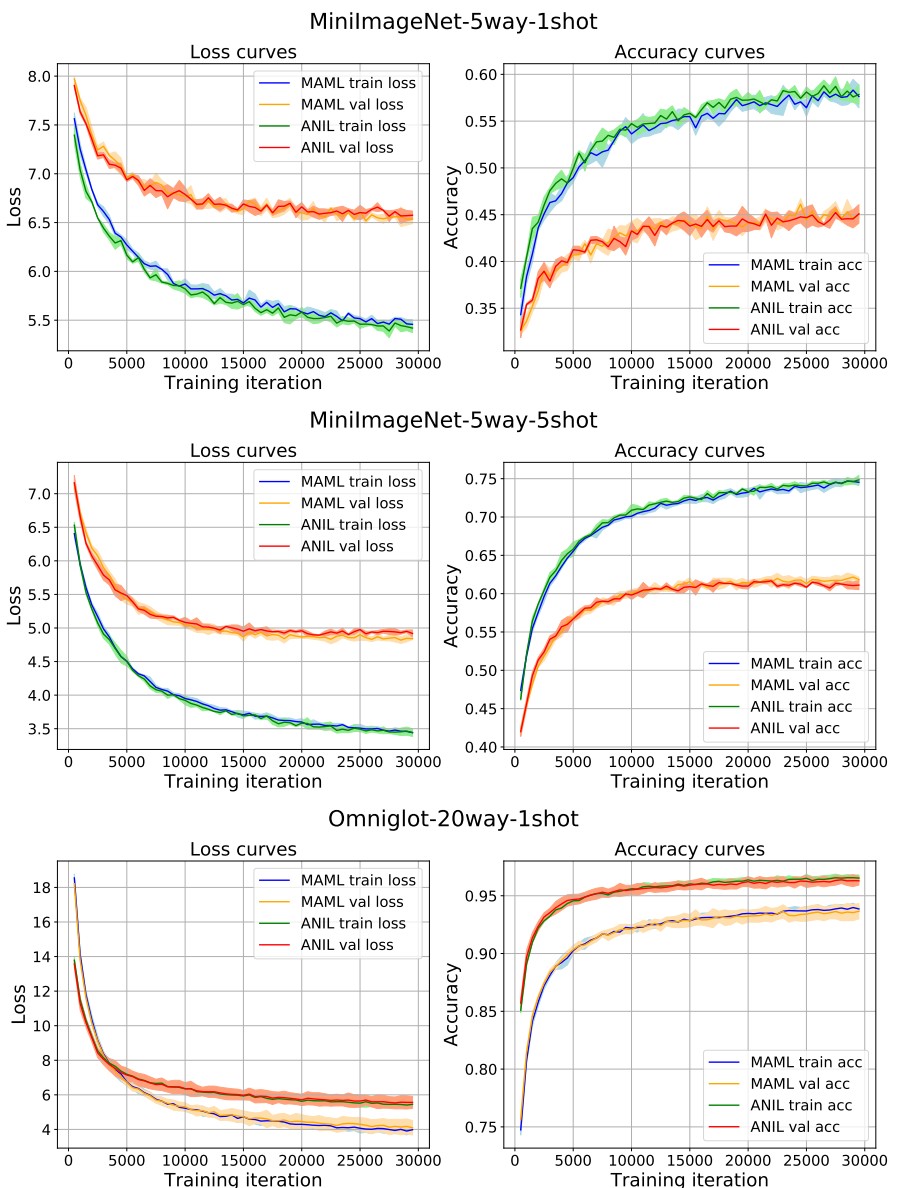

Figure 10: **ANIL and MAML on MiniImageNet and Omniglot**. Loss and accuracy curves for ANIL and MAML on (i) MiniImageNet-5way-1shot (ii) MiniImageNet-5way-5shot (iii) Omniglot-20way-1shot. These illustrate how both algorithms learn very similarly over training.

implementation from the original authors. We note that the results for MAML in these RL domains do not exactly match those in the original paper; this may be due to large variance in results, depending on the random initialization. We used the same model architecture as the original paper (two layer MLP with 100 hidden units in each layer), a batch size of 40, 1 inner loop update step with an inner learning rate of 0.1 and 20 trajectories for inner loop adaptation. We trained three MAML and ANIL models with different random initialization, and quote the mean and standard deviation of the results. As in the original MAML paper, for RL experiments, we select the best performing model over 500 iterations of training and evaluate this model at test time on a new set of tasks.

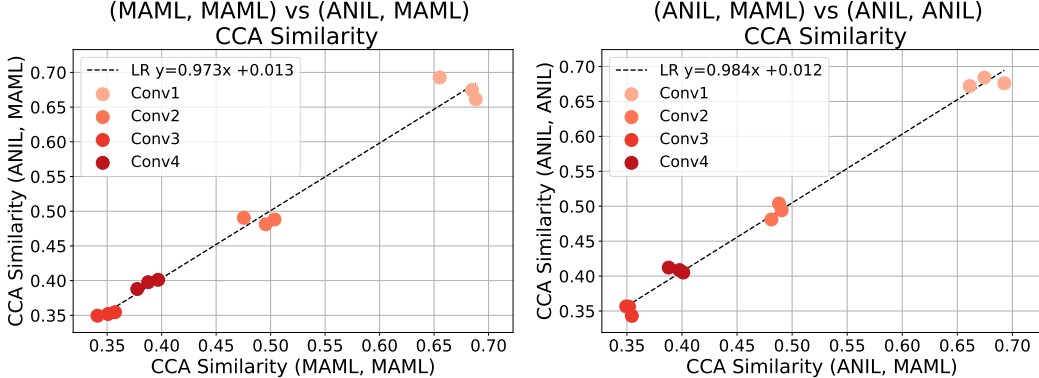

Figure 11: **Computing CCA similarity across different seeds of MAML and ANIL networks suggests these representations are similar.** We plot the CCA similarity between an ANIL seed and a MAML seed, plotted against (i) the MAML seed compared to a different MAML seed (ii) the ANIL seed compared to a different ANIL seed. We observe a strong correlation of similarity scores in both (i) and (ii). This tells us that (i) two MAML representations vary about as much as MAML and ANIL representations (ii) two ANIL representations vary about as much as MAML and ANIL representations. In particular, this suggests that MAML and ANIL learn similar features, despite having significant algorithmic differences.

|  | Training: 5way-1shot | | | Training: 5way-5shot | | |
|---|---|---|---|---|---|---|
|  | Mean (s) | Median (s) | Speedup | Mean (s) | Median (s) | Speedup |
| MAML | 0.15 | 0.13 | 1 | 0.68 | 0.67 | 1 |
| First Order MAML | 0.089 | 0.083 | 1.69 | 0.40 | 0.39 | 1.7 |
| ANIL | 0.084 | 0.072 | 1.79 | 0.37 | 0.36 | 1.84 |

|  | Inference: 5way-1shot | | | Inference: 5way-5shot | | |
|---|---|---|---|---|---|---|
|  | Mean (s) | Median (s) | Speedup | Mean (s) | Median (s) | Speedup |
| MAML | 0.083 | 0.078 | 1 | 0.37 | 0.36 | 1 |
| ANIL | 0.020 | 0.017 | 4.15 | 0.076 | 0.071 | 4.87 |

Table 6: **ANIL offers significant computational speedup over MAML, during both training and inference.** Table comparing execution times and speedups of MAML, First Order MAML, and ANIL during training (above) and inference (below) on MiniImageNet domains. Speedup is calculated relative to MAML's execution time. We see that ANIL offers noticeable speedup over MAML, as a result of removing the inner loop almost completely. This permits faster training and inference.

## C.5 ANIL IS COMPUTATIONALLY SIMPLER THAN MAML

Table 6 shows results from a comparison of the computation time for MAML, First Order MAML, and ANIL, during training and inference, with the TensorFlow implementation described previously, on both MiniImageNet domains. These results are average time for executing forward and backward passes during training (above) and a forward pass during inference (bottom), for a task batch size of 1, and a target set size of 1. Results are averaged over 2000 such batches. Speedup is calculated relative to MAML's execution time. Each batches' images were loaded into memory before running the TensorFlow computation graph, to ensure that data loading time was not captured in the timing. Experiments were run on a single NVIDIA Titan-Xp GPU.

During training, we see that ANIL is as fast as First Order MAML (which does not compute second order terms during training), and about 1.7x as fast as MAML. This leads to a significant overall training speedup, especially when coupled with the fact that the rate of learning for these ANIL and MAML is very similar; see learning curves in Appendix C.2. Note that unlike First Order MAML, ANIL also performs very comparably to MAML on benchmark tasks (on some tasks, First Order MAML performs worse (Finn et al., 2017)). During inference, ANIL achieves over a 4x speedup over MAML (and thus also 4x over First Order MAML, which is identical to MAML at inference

| Method | MiniImageNet-5way-1shot | MiniImageNet-5way-5shot |
|---|---|---|
| MAML training-MAML head | $46.9 \pm 0.2$ | $63.1 \pm 0.4$ |
| MAML training-NIL head | $48.4 \pm 0.3$ | $61.5 \pm 0.8$ |
| ANIL training-ANIL head | $46.7 \pm 0.4$ | $61.5 \pm 0.5$ |
| ANIL training-NIL head | $48.0 \pm 0.7$ | $62.2 \pm 0.5$ |
| Multiclass pretrain-MAML head | $38.4 \pm 0.8$ | $54.6 \pm 0.4$ |
| Multiclass pretrain-NIL head | $39.7 \pm 0.3$ | $54.4 \pm 0.5$ |
| Multitask pretrain-MAML head | $26.5 \pm 0.8$ | $32.8 \pm 0.6$ |
| Multitask pretrain-NIL head | $26.5 \pm 1.1$ | $34.2 \pm 3.5$ |
| Random features-MAML head | $32.1 \pm 0.5$ | $43.1 \pm 0.3$ |
| Random features-NIL head | $32.9 \pm 0.6$ | $43.2 \pm 0.5$ |

Table 7: **Test time performance is dominated by features learned, with no difference between NIL/MAML heads.** We see identical performances of MAML/NIL heads at test time, indicating that MAML/ANIL training leads to better learned features.

time). Both training and inference speedups illustrate the significant computational benefit of ANIL over MAML.

# D FURTHER RESULTS ON THE NETWORK HEAD AND BODY

## D.1 TRAINING REGIMES FOR THE NETWORK BODY

We add to the results of Section 5.2 in the main text by seeing if training a head and applying that to the representations at test time (instead of the NIL algorithm) gives in any change in the results. As might be predicted by Section 5.1, we find no change the results.

More specifically, we do the following:

- We train MAML/ANIL networks as standard, and do standard test time adaptation.
- For multiclass training, we first (pre)train with multiclass classification, then throw away the head and freeze the body. We initialize a new e.g. 5-class head, and train that (on top of the frozen multiclass pretrained features) with MAML. At test time we perform standard adaptation.
- The same process is applied to multitask training.
- A similar process is applied to random features, except the network is initialized and then frozen.

The results of this, along with the results from Table 5 in the main text is shown in Table 7. We observe very little performance difference between using a MAML/ANIL head and a NIL head for each training regime. Specifically, task performance is purely determined by the quality of the features and representations learned during training, with task-specific alignment at test time being (i) unnecessary (ii) unable to influence the final performance of the model (e.g. multitask training performance is equally with a MAML head as it is with a NIL-head.)

## D.2 REPRESENTATIONAL ANALYSIS OF DIFFERENT TRAINING REGIMES

In Table 8 we include results on using CCA and CKA on the representations learned by the different training methods. Specifically, we studied how similar representations of different training methods were to MAML training, finding a direct correlation with performance – training schemes learning representations most similar to MAML also performed the best. We computed similarity scores by averaging the scores over the first three conv layers in the body of the network.

| Feature pair | CCA Similarity | CKA Similarity |
|---|---|---|
| (MAML, MAML) | 0.51 | 0.83 |
| (Multiclass pretrain, MAML) | 0.48 | 0.79 |
| (Random features, MAML) | 0.40 | 0.72 |
| (Multitask pretrain, MAML) | 0.28 | 0.65 |

Table 8: **MAML training most closely resembles multiclass pretraining, as illustrated by CCA and CKA similarities**. On analyzing the CCA and CKA similarities between different baseline models and MAML (comparing across different tasks and seeds), we see that multiclass pretraining results in features most similar to MAML training. Multitask pretraining differs quite significantly from MAML-learned features, potentially due to the alignment problem.

