# OpenReview forum: "Rapid Learning or Feature Reuse? Towards Understanding the Effectiveness of MAML"
_ICLR.cc/2020/Conference — Accept (Poster)_

### Official Review · AnonReviewer2 · 2019-10-15
**Official Blind Review #2**

**Rating:** 8

**Review:**

The paper claims to examine the reasons for the success of MAML---an influential meta-learning algorithm to tackle few-shot learning. It thoroughly investigated the importance of the two optimization loops, and found that feature reuse is the dominant factor for MAML’s success. Moreover, the authors proposed new algorithms---ANIL and NIL---which spend much less computation on the inner loop of MAML. They also discussed their findings in a broader meta-learning context.

I think the paper should be accepted for the following reasons:

1. The experimental study is thorough.

The experiments follow a rigorous design of hypothesis-checking style and the conclusions are supported by extensive results under various evaluations.

The findings are potentially helpful for many future works in this field.

2. The paper is clearly written.

It is generally enjoyable to read, except for some minor things to improve: (1) Evaluation metrics in table-2, table-4 and table-5 had better be explicitly clarified in the captions (2) No subsection seems needed in section-6.


**Experience Assessment:**

I do not know much about this area.

**Review Assessment: Checking Correctness Of Derivations And Theory:**

N/A

**Review Assessment: Checking Correctness Of Experiments:**

I carefully checked the experiments.

**Review Assessment: Thoroughness In Paper Reading:**

I read the paper thoroughly.

---

> ### Author Response · Authors · 2019-11-08
> **Author Response to Review #2**
>
> Thank you for the thorough reading of our paper and the positive feedback. We will add these edits you suggested in the next revision!
>
> Since submission we have also added additional interesting experiments to test how important the features that layers learn in the meta-initialization is to performance. We do this by resetting contiguous blocks of layers at meta-initialization to the very start of training, finding that resetting the lowest layer leads to the greatest performance drop. These experiments further support the feature reuse paradigm and also show that features in the earliest layers are most important for good performance:
>
> __________________________________________________________________
> Layers Reset           MiniImageNet-5way-1shot               MiniImageNet-5way-5shot
> __________________________________________________________________
>   None                                  46.7                                                  61.5
>   1                                         31.3                                                  39.4
>   1,2                                      28.8                                                  36.8
>   1,2,3                                   29.3                                                  37.3
>   1,2,3,4                                27.5                                                  35.9

---

### Official Review · AnonReviewer1 · 2019-10-23
**Official Blind Review #1**

**Rating:** 3

**Review:**

This paper is exploring the importance of the inner loop in MAML. It shows that using the inner loop only for the classifier head (ANIL) results are comparable to MAML. It also shows that using no inner loop at all (NIL) is okay for test time but not for training time.

It is indeed interesting to understand the effect of the inner loop. But, as the authors noted (“Our work is complementary to methods extending MAML, and our simplification and insights could be applied to such extensions also”), for it to be useful I’d like to see whether these insights can be extended to SOTA models. MAML is less than 50% accuracy on 1-shot mini-imagenet while current SOTS models achieve 60-65%.

The NIL experiment that shows low performance when no inner loop is used in training time doesn’t make sense. This is basically the same as the nearest-neighbour family of methods, e.g. ProtoNet (Snell et al., 2017), which have been shown to perform similarly to (or even better than) MAML.


After rebuttal:
I do think it's important to also have that kind of analysis works. My main concern is with how ANIL and NIL are introduced as new algorithms and not just an ablation of the MAML method. Presented as new algorithms I tend to compare them against the leaderboard where they are very far from the top. I am keeping my previous rating.

**Experience Assessment:**

I have published one or two papers in this area.

**Review Assessment: Checking Correctness Of Derivations And Theory:**

I assessed the sensibility of the derivations and theory.

**Review Assessment: Checking Correctness Of Experiments:**

I assessed the sensibility of the experiments.

**Review Assessment: Thoroughness In Paper Reading:**

I read the paper at least twice and used my best judgement in assessing the paper.

---

> ### Author Response · Authors · 2019-11-08
> **Author Response to Review #1**
>
> Thank you for your review. We respectfully disagree with your overall assessment of our paper, and offer here some justification.
>
> Firstly, the focus of this paper is understanding why the highly popular and influential MAML algorithm works, and *not* just about removing the inner loop. The key message is that somewhat surprisingly, there is significant feature reuse, and not much large adaptation.  ANIL and NIL arise as a consequence of this, providing useful insights into further understanding the meta-learning field.
>
> In line with the goal of gaining *fundamental insights* into these algorithms, the scope of this paper was not to look at SOTA methods, but perform a thorough and detailed study of the main MAML algorithm, and have **entirely reproducible** experiments. To do this, we stuck to using the following standard, easy to use and open source repositories:
> https://github.com/cbfinn/maml for the few shot classification results (Code for the few-shot learning experiments from the original MAML paper, open-sourced by the original authors).
> https://github.com/tristandeleu/pytorch-maml-rl for RL experiments (Code to reproduce the RL results from the original MAML paper).
>
> We show how these insights on feature reuse connect to how other metalearning algorithms e.g. (Matching Networks, Vinyals et al) work, in Section 6. Note that feature reuse is a key factor behind the performance of other metalearning methods that achieve near SOTA , e.g. (Meta-learning with differentiable convex optimization, Kwonjoon et al), which does no ‘adaptation’ at test time, and still achieves excellent performance on the benchmarks. By demonstrating the importance of feature reuse in MAML (which is not at all immediately evident, due to the inner loop optimization), and identifying that other metalearning methods also employ feature reuse effectively, we link understanding about how MAML works to how other algorithms work. This contribution goes beyond just removing the inner loop in MAML.
>
> Regarding NIL at training time and ProtoNets: firstly, we again emphasize that NIL at training time was **not at all** the main focus of the paper. Additionally, there are **several very important differences** between NIL and Protonet:
> — ProtoNets compute prototypes (averages over representations from the same class in the support set), which we do not do, taking the raw cosine similarity between test examples and all support set examples.
> — ProtoNets use euclidean distance, instead of cosine distance, which is explicitly stated in the paper to improve performance.
> — ProtoNets use a *learning rate scheduler* which also helps performance, while for using NIL during training we simply use the Adam optimizer with the default settings used in the original MAML paper, as we are seeking to compare NIL as closely as possible with the original MAML paper, not search over potential optimization curricula to improve its performance.
> — ProtoNets have a different training process, with a *larger* number of classes during training compared to testing. E.g for testing 5-shot performance, they perform 20-way classification during the meta-training stage.
> These key differences make it very hard to compare performance when training with NIL to  performance from ProtoNets.
>
> Since submission we have also added additional interesting experiments to test how important the features learned in the meta-initialization are. We reset contiguous blocks of layers at meta-initialization to the very start of training. These experiments further support the feature reuse paradigm and also show that features in the earliest layers are most important for good performance:
> __________________________________________________________________
> Layers Reset           MiniImageNet-5way-1shot               MiniImageNet-5way-5shot
> __________________________________________________________________
>   None                                  46.7                                                  61.5
>   1                                         31.3                                                  39.4
>   1,2                                      28.8                                                  36.8
>   1,2,3                                   29.3                                                  37.3
>   1,2,3,4                                27.5                                                  35.9
>
> The MAML algorithm has been extensively developed in recent literature, and has been applied to a wide range of problems. Our work offers fundamental insight into why this highly popular algorithm is effective, thereby answering a central open question. In investigating this question, we conducted detailed, reproducible experiments, that are of interest to the community, as seen by public comments. Our paper provides an excellent foundation for future work, which could leverage our insights to both understand few-shot learning algorithms better and develop improved few-shot learning algorithms.

---

### Official Review · AnonReviewer4 · 2019-11-15
**Official Blind Review #4**

**Rating:** 8

**Review:**

After  rebuttal period: I recommend accepting  this  paper.
======================================
Summary:

This paper attempts to understand if the success of MAML is due to rapid learning or feature reuse. The analysis shows that MAML is performing better mainly due to feature reuse. Authors use this result to derive a simpler version of MAML called ANIL. ANIL does not update the non-final layers of the network during inner loop training and still has similar performance to MAML.

My comments:

Overall I think this is an interesting analysis paper which sheds some light on how MAML works, However, I see these analysis not just as a criticism towards MAML. I also see these analysis as a criticism against the meta-learning datasets that we use. All these datasets are artifically created from the same dataset and hence it might be very easy to reuse features to get good performance. I am not sure if the same analysis will hold if we consider a dataset where tasks are not this similar (like Meta-dataset, Triantafillou et al 2019). I encourage the authors to have this disclaimer in the end of the paper so that the community does not falsely conclude that MAML cannot do rapid learning.

**Experience Assessment:**

I have read many papers in this area.

**Review Assessment: Checking Correctness Of Derivations And Theory:**

N/A

**Review Assessment: Checking Correctness Of Experiments:**

I assessed the sensibility of the experiments.

**Review Assessment: Thoroughness In Paper Reading:**

I made a quick assessment of this paper.

---

> ### Author Response · Authors · 2019-11-15
> **Author Response to Review #4**
>
> Thank you very much for your review and comments on our paper. We will update the latest version of our paper with the disclaimer you have mentioned. We think that exploring how our analysis applies to other datasets from Meta-dataset (Triantafillou et al 2019), or across more diverse tasks, will be interesting future work.

---

### Public Comment · ~Luisa_M_Zintgraf1 · 2019-10-14
**Clarification of Related Work**


Dear authors,

I read your paper with great interest and am excited to see more insights into understanding how gradient-based meta-learning methods work. Our work [1] from this year’s ICRL comes to similar conclusions. I wanted to take this opportunity to clarify our work, since this is currently not represented accurately in your paper.

In your paper you say that [1] “has looked at having outer/inner loop specific parameters, but does this in a more complex fashion, partitioning parameters within each layer, and for specific layers, contrasting with the simple head/body separation in ANIL”. That’s not true. We do not partition the parameters within each layer, and there is only a single (additional input) vector which is updated in the inner loop. So in fact, we also do a very simple partition, which could be described as a tail/body separation compared to the body/head separation of ANIL.

In short: Our method CAVIA separates the network into two parts, a single input vector (“tail” or“context parameter”) and the main network (“body”). In the inner loop we only adapt the context parameter, and in the outer loop we adapt the main body of the network. The intuition behind this idea is that for many tasks, if we had access to a true task description (e.g., in the HalfCheetah environments the direction/velocity) we could just condition the network on this and there would be no need for meta-learning. See also the simplified architecture visualisation in our blog post: http://whirl.cs.ox.ac.uk/blog/cavia/.

Can you please rectify the description of our work? Please do let me know if you have any questions.

[1] Fast context adaptation via meta-learning. Luisa Zintgraf, Kyriacos Shiarlis, Vitaly Kurin, Katja Hofmann, Shimon Whiteson. ICLR 2019.

---

> ### Author Response · Authors · 2019-10-15
> **Thank you for the comments on related work -- we have updated our description accordingly.**
>
> Thank you for your interest in our paper, the helpful outline of CAVIA, and the pointer to the blog post. We have updated the description of CAVIA in a newer version of our paper.
>
> We feel that there are exciting open directions to explore relating the context parameter-main network split in CAVIA and the head-body split we analyze; for example, how adding a context vector to hidden layer(s) affects what is learned by the head of the network, and whether there is a variant of the NIL algorithm that can incorporate the context vector. From an analysis point of view, we also think it is interesting to understand the representations learned by the higher dimensional context vectors using analysis tools such as CCA/CKA (as in our paper).

---

### Public Comment · ~Khurram_Javed1 · 2019-11-01
**Some recent related work**

Thank you for this nice paper. I really enjoyed reading it.

I just wanted to point to our work -- Javed and Martha, 2019 [1] -- which proposes a meta-learning framework similar to ANIL. More concretely, ANIL is a special case of our method (See Figure 1 in the paper: https://arxiv.org/pdf/1905.12588.pdf). For a linear RLN, our method is equivalent to ANIL (In our experiments, however, we used a single hidden layer RLN).

I want to be clear to the reviewers that I'm not suggesting this paper is not adding to the meta-learning discourse. Our work and this paper -- even though propose a similar solution method -- look at the problem from a very different perspective. This paper shows that even when the meta-parameters represent a model initialization, MAML is essentially doing feature reuse (in other words, it is just learning a representation which allows for quick adaptation).

We, on the other hand, directly define our meta-parameters as a representation learning network; we DO NOT investigate if MAML is already implicitly doing this (which is the main contribution of this paper). In fact, the results in this paper came as a surprise to me and added to my understanding of meta-learners that learn a model initialization.

[1] Javed and White, Meta-Learning Representations for Continual Learning, NeurIPS19.
Link: https://arxiv.org/pdf/1905.12588.pdf

---

> ### Author Response · Authors · 2019-11-08
> **Thank you for your interest and the reference!**
>
> Thank you very much for your comment and for highlighting this related work -- it is very interesting to see the parallels between the method you have proposed and the ANIL algorithm, especially seeing how ANIL-like ideas apply to continual learning and catastrophic forgetting. We will include a reference to this work in the updated version of our paper.

---

### Decision · Program_Chairs · 2019-12-19

**Decision:**

Accept (Poster)

**Comment:**

Paper received mixed reviews: WR (R1), A (R2 and R3). AC has read reviews/rebuttal and examined paper. AC agrees that R1's concerns are misplaced and feels the paper should be accepted.